# Discrimination-aware Channel Pruning
# for Deep Neural Networks

**Zhuangwei Zhuang**[1*], **Mingkui Tan**[1*†], **Bohan Zhuang**[2*], **Jing Liu**[1*],
**Yong Guo**[1], **Qingyao Wu**[1], **Junzhou Huang**[3,4], **Jinhui Zhu**[1†]
[1]South China University of Technology, [2]The University of Adelaide,
[3]University of Texas at Arlington, [4]Tencent AI Lab
{z.zhuangwei, seliujing, guo.yong}@mail.scut.edu.cn, jzhuang@uta.edu
{mingkuitan, qyw, csjhzhu}@scut.edu.cn, bohan.zhuang@adelaide.edu.au

## Abstract

Channel pruning is one of the predominant approaches for deep model compression. Existing pruning methods either train from scratch with sparsity constraints on channels, or minimize the reconstruction error between the pre-trained feature maps and the compressed ones. Both strategies suffer from some limitations: the former kind is computationally expensive and difficult to converge, whilst the latter kind optimizes the reconstruction error but ignores the discriminative power of channels. In this paper, we investigate a simple-yet-effective method called discrimination-aware channel pruning (DCP) to choose those channels that really contribute to discriminative power. To this end, we introduce additional discrimination-aware losses into the network to increase the discriminative power of intermediate layers and then select the most discriminative channels for each layer by considering the additional loss and the reconstruction error. Last, we propose a greedy algorithm to conduct channel selection and parameter optimization in an iterative way. Extensive experiments demonstrate the effectiveness of our method. For example, on ILSVRC-12, our pruned ResNet-50 with 30% reduction of channels outperforms the baseline model by 0.39% in top-1 accuracy.

## 1 Introduction

Since 2012, convolutional neural networks (CNNs) have achieved great success in many computer vision tasks, e.g., image classification [21, 41], face recognition [37, 42], object detection [35, 36], image generation [7, 3] and video analysis [38, 47]. However, deep models are often with a huge number of parameters and the model size is very large, which incurs not only huge memory requirement but also unbearable computation burden. As a result, deep learning methods are hard to be applied on hardware devices with limited storage and computation resources, such as cell phones. To address this problem, model compression is an effective approach, which aims to reduce the model redundancy without significant degeneration in performance.

Recent studies on model compression mainly contain three categories, namely, quantization [34, 54], sparse or low-rank compressions [10, 11], and channel pruning [27, 28, 51, 49]. Network quantization seeks to reduce the model size by quantizing float weights into low-bit weights (e.g., 8 bits or even 1 bit). However, the training is very difficult due to the introduction of quantization errors. Making sparse connections can reach high compression rate in theory, but it may generate irregular convolutional kernels which need sparse matrix operations for accelerating the computation. In

---

contrast, channel pruning reduces the model size and speeds up the inference by removing redundant channels directly, thus little additional effort is required for fast inference. On top of channel pruning, other compression methods such as quantization can be applied. In fact, pruning redundant channels often helps to improve the efficiency of quantization and achieve more compact models.

Identifying the informative (or important) channels, also known as channel selection, is a key issue in channel pruning. Existing works have exploited two strategies, namely, training-from-scratch methods which directly learn the importance of channels with sparsity regularization [1, 27, 48], and reconstruction-based methods [14, 16, 24, 28]. Training-from-scratch is very difficult to train especially for very deep networks on large-scale datasets. Reconstruction-based methods seek to do channel pruning by minimizing the reconstruction error of feature maps between the pruned model and a pre-trained model [14, 28]. These methods suffer from a critical limitation: an actually redundant channel would be mistakenly kept to minimize the reconstruction error of feature maps. Consequently, these methods may result in apparent drop in accuracy on more compact and deeper models such as ResNet [13] for large-scale datasets.

In this paper, we aim to overcome the drawbacks of both strategies. First, in contrast to existing methods [14, 16, 24, 28], we assume and highlight that an informative channel, no matter where it is, should own discriminative power; otherwise it should be deleted. Based on this intuition, we propose to find the channels with true discriminative power for the network. Specifically, relying on a pre-trained model, we add multiple additional losses (i.e., discrimination-aware losses) evenly to the network. For each stage, we first do fine-tuning using one additional loss and the final loss to improve the discriminative power of intermediate layers. And then, we conduct channel pruning for each layer involved in the considered stage by considering both the additional loss and the reconstruction error of feature maps. In this way, we are able to make a balance between the discriminative power of channels and the feature map reconstruction.

Our main contributions are summarized as follows. First, we propose a discrimination-aware channel pruning (DCP) scheme for compressing deep models with the introduction of additional losses. DCP is able to find the channels with true discriminative power. DCP prunes and updates the model stage-wisely using a proper discrimination-aware loss and the final loss. As a result, it is not sensitive to the initial pre-trained model. Second, we formulate the channel selection problem as an $\ell_{2,0}$-norm constrained optimization problem and propose a greedy method to solve the resultant optimization problem. Extensive experiments demonstrate the superior performance of our method, especially on deep ResNet. On ILSVRC-12 [4], when pruning 30% channels of ResNet-50, DCP improves the original ResNet model by 0.39% in top-1 accuracy. Moreover, when pruning 50% channels of ResNet-50, DCP outperforms ThiNet [28], a state-of-the-art method, by 0.81% and 0.51% in top-1 and top-5 accuracy, respectively.

## 2  Related studies

**Network quantization.**  In [34], Rastegari *et al.* propose to quantize parameters in the network into $+1/-1$. The proposed BWN and XNOR-Net can achieve comparable accuracy to their full-precision counterparts on large-scale datasets. In [55], high precision weights, activations and gradients in CNNs are quantized to low bit-width version, which brings great benefits for reducing resource requirement and power consumption in hardware devices. By introducing zero as the third quantized value, ternary weight networks (TWNs) [23, 56] can achieve higher accuracy than binary neural networks. Explorations on quantization [54, 57] show that quantized networks can even outperform the full precision networks when quantized to the values with more bits, e.g., 4 or 5 bits.

**Sparse or low-rank connections.**  To reduce the storage requirements of neural networks, Han *et al.* suggest that neurons with zero input or output connections can be safely removed from the network [12]. With the help of the $\ell_1/\ell_2$ regularization, weights are pushed to zeros during training. Subsequently, the compression rate of AlexNet can reach $35\times$ with the combination of pruning, quantization, and Huffman coding [11]. Considering the importance of parameters is changed during weight pruning, Guo *et al.* propose dynamic network surgery (DNS) in [10]. Training with sparsity constraints [40, 48] has also been studied to reach higher compression rate.

Deep models often contain a lot of correlations among channels. To remove such redundancy, low-rank approximation approaches have been widely studied [5, 6, 19, 39]. For example, Zhang *et*

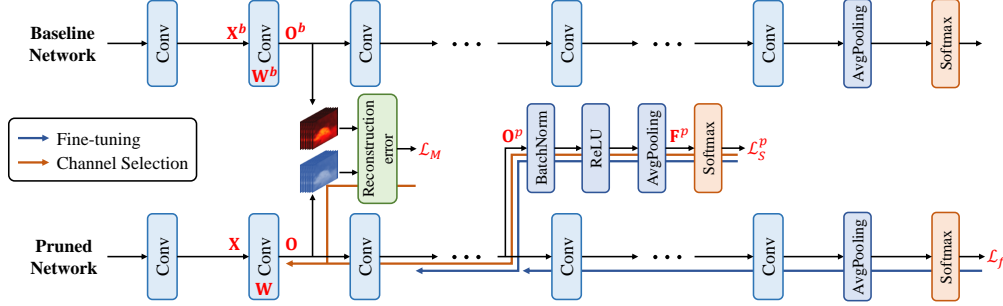

Figure 1: Illustration of discrimination-aware channel pruning. Here, $\mathcal{L}_S^p$ denotes the discrimination-aware loss (e.g., cross-entropy loss) in the $L_p$-th layer, $\mathcal{L}_M$ denotes the reconstruction loss, and $\mathcal{L}_f$ denotes the final loss. For the $p$-th stage, we first fine-tune the pruned model by $\mathcal{L}_S^p$ and $\mathcal{L}_f$, then conduct the channel selection for each layer in $\{L_{p-1} + 1, \ldots, L_p\}$ with $\mathcal{L}_S^p$ and $\mathcal{L}_M$.

*al.* speed up VGG for $4\times$ with negligible performance degradation on ImageNet [53]. However, low-rank approximation approaches are unable to remove those redundant channels that do not contribute to the discriminative power of the network.

**Channel pruning.** Compared with network quantization and sparse connections, channel pruning removes both channels and the related filters from the network. Therefore, it can be well supported by existing deep learning libraries with little additional effort. The key issue of channel pruning is to evaluate the importance of channels. Li *et al.* measure the importance of channels by calculating the sum of absolute values of weights [24]. Hu *et al.* define average percentage of zeros (APoZ) to measure the activation of neurons [16]. Neurons with higher values of APoZ are considered more redundant in the network. With a sparsity regularizer in the objective function, training-based methods [1, 27] are proposed to learn the compact models in the training phase. With the consideration of efficiency, reconstruction-methods [14, 28] transform the channel selection problem into the optimization of reconstruction error and solve it by a greedy algorithm or LASSO regression.

## 3 Proposed method

Let $\{\mathbf{x}_i, y_i\}_{i=1}^N$ be the training samples, where $N$ indicates the number of samples. Given an $L$-layer CNN model $M$, let $\mathbf{W} \in \mathbb{R}^{n \times c \times h_f \times z_f}$ be the model parameters w.r.t. the $l$-th convolutional layer (or block), as shown in Figure 1. Here, $h_f$ and $z_f$ denote the height and width of filters, respectively; $c$ and $n$ denote the number of **input** and **output** channels, respectively. For convenience, hereafter we omit the layer index $l$. Let $\mathbf{X} \in \mathbb{R}^{N \times c \times h_{in} \times z_{in}}$ and $\mathbf{O} \in \mathbb{R}^{N \times n \times h_{out} \times z_{out}}$ be the input feature maps and the involved output feature maps, respectively. Here, $h_{in}$ and $z_{in}$ denote the height and width of the input feature maps, respectively; $h_{out}$ and $z_{out}$ represent the height and width of the output feature maps, respectively. Moreover, let $\mathbf{X}_{i,k,:,:}$ be the feature map of the $k$-th channel for the $i$-th sample. $\mathbf{W}_{j,k,:,:}$ denotes the parameters w.r.t. the $k$-th input channel and $j$-th output channel. The output feature map of the $j$-th channel for the $i$-th sample, denoted by $\mathbf{O}_{i,j,:,:}$, is computed by

$$\mathbf{O}_{i,j,:,:} = \sum_{k=1}^c \mathbf{X}_{i,k,:,:} * \mathbf{W}_{j,k,:,:}, \tag{1}$$

where $*$ denotes the convolutional operation.

Given a pre-trained model $M$, the task of **Channel Pruning** is to prune those redundant channels in $\mathbf{W}$ to save the model size and accelerate the inference speed in Eq. (1). In order to choose channels, we introduce a variant of $\ell_{2,0}$-norm $||\mathbf{W}||_{2,0} = \sum_{k=1}^c \Omega(\sum_{j=1}^n ||\mathbf{W}_{j,k,:,:}||_F)$, where $\Omega(a) = 1$ if $a \neq 0$ and $\Omega(a) = 0$ if $a = 0$, and $|| \cdot ||_F$ represents the Frobenius norm. To induce sparsity, we can impose an $\ell_{2,0}$-norm constraint on $\mathbf{W}$:

$$||\mathbf{W}||_{2,0} = \sum_{k=1}^c \Omega(\sum_{j=1}^n ||\mathbf{W}_{j,k,:,:}||_F) \leq \kappa_l, \tag{2}$$

where $\kappa_l$ denotes the desired number of channels at the layer $l$. Or equivalently, given a predefined pruning rate $\eta \in (0, 1)$ [1, 27], it follows that $\kappa_l = \lceil \eta c \rceil$.

## 3.1 Motivations

Given a pre-trained model $M$, existing methods [14, 28] conduct channel pruning by minimizing the reconstruction error of feature maps between the pre-trained model $M$ and the pruned one. Formally, the reconstruction error can be measured by the mean squared error (MSE) between feature maps of the baseline network and the pruned one as follows:

$$\mathcal{L}_M(\mathbf{W}) = \frac{1}{2Q} \sum_{i=1}^{N} \sum_{j=1}^{n} ||\mathbf{O}_{i,j,:,:}^b - \mathbf{O}_{i,j,:,:}||_F^2, \qquad (3)$$

where $Q = N \cdot n \cdot h_{out} \cdot z_{out}$ and $\mathbf{O}_{i,j,:,:}^b$ denotes the feature maps of the baseline network. Reconstructing feature maps can preserve most information in the learned model, but it has two limitations. First, the pruning performance is highly affected by the quality of the pre-trained model $M$. If the baseline model is not well trained, the pruning performance can be very limited. Second, to achieve the minimal reconstruction error, some channels in intermediate layers may be mistakenly kept, even though they are actually not relevant to the discriminative power of the network. This issue will be even severer when the network becomes deeper.

In this paper, we seek to do channel pruning by keeping those channels that really contribute to the discriminative power of the network. In practice, however, it is very hard to measure the discriminative power of channels due to the complex operations (such as ReLU activation and Batch Normalization) in CNNs. One may consider one channel as an important one if the final loss $\mathcal{L}_f$ would sharply increase without it. However, it is not practical when the network is very deep. In fact, for deep models, its shallow layers often have little discriminative power due to the long path of propagation.

To increase the discriminative power of intermediate layers, one can introduce additional losses to the intermediate layers of the deep networks [43, 22, 8]. In this paper, we insert $P$ discrimination-aware losses $\{\mathcal{L}_S^p\}_{p=1}^P$ evenly into the network, as shown in Figure 1. Let $\{L_1, ..., L_P, L_{P+1}\}$ be the layers at which we put the losses, with $L_{P+1} = L$ being the final layer. For the $p$-th loss $\mathcal{L}_S^p$, we consider doing channel pruning for layers $l \in \{L_{p-1} + 1, ..., L_p\}$, where $L_{p-1} = 0$ if $p = 1$. It is worth mentioning that, we can add one loss to each layer of the network, where we have $L_l = l$. However, this can be very computationally expensive yet not necessary.

## 3.2 Construction of discrimination-aware loss

The construction of discrimination-aware loss $\mathcal{L}_S^p$ is very important in our method. As shown in Figure 1, each loss uses the output of layer $L_p$ as the input feature maps. To make the computation of the loss feasible, we impose an average pooling operation over the feature maps. Moreover, to accelerate the convergence, we shall apply batch normalization [18, 9] and ReLU [29] before doing the average pooling. In this way, the input feature maps for the loss at layer $L_p$, denoted by $\mathbf{F}^p(\mathbf{W})$, can be computed by

$$\mathbf{F}^p(\mathbf{W}) = \text{AvgPooling}(\text{ReLU}(\text{BN}(\mathbf{O}^p))), \qquad (4)$$

where $\mathbf{O}^p$ represents the output feature maps of layer $L_p$. Let $\mathbf{F}^{(p,i)}$ be the feature maps w.r.t. the $i$-th example. The discrimination-aware loss w.r.t. the $p$-th loss is formulated as

$$\mathcal{L}_S^p(\mathbf{W}) = -\frac{1}{N} \left[ \sum_{i=1}^{N} \sum_{t=1}^{m} I\{y^{(i)} = t\} \log \frac{e^{\boldsymbol{\theta}_t^\top \mathbf{F}^{(p,i)}}}{\sum_{k=1}^{m} e^{\boldsymbol{\theta}_k^\top \mathbf{F}^{(p,i)}}} \right], \qquad (5)$$

where $I\{\cdot\}$ is the indicator function, $\boldsymbol{\theta} \in \mathbb{R}^{n_p \times m}$ denotes the classifier weights of the fully connected layer, $n_p$ denotes the number of input channels of the fully connected layer and $m$ is the number of classes. Note that we can use other losses such as angular softmax loss [26] as the additional loss.

In practice, since a pre-trained model contains very rich information about the learning task, similar to [28], we also hope to reconstruct the feature maps in the pre-trained model. By considering both cross-entropy loss and reconstruction error, we have a joint loss function as follows:

$$\mathcal{L}(\mathbf{W}) = \mathcal{L}_M(\mathbf{W}) + \lambda \mathcal{L}_S^p(\mathbf{W}), \qquad (6)$$

where $\lambda$ balances the two terms.

**Proposition 1 (Convexity of the loss function)** *Let $\mathbf{W}$ be the model parameters of a considered layer. Given the mean square loss and the cross-entropy loss defined in Eqs. (3) and (5), then the joint loss function $\mathcal{L}(\mathbf{W})$ is convex w.r.t. $\mathbf{W}$.*[3]

Last, the optimization problem for discrimination-aware channel pruning can be formulated as

$$\min_{\mathbf{W}} \quad \mathcal{L}(\mathbf{W}), \quad \text{s.t. } ||\mathbf{W}||_{2,0} \leq \kappa_l, \tag{7}$$

where $\kappa_l < c$ is the number channels to be selected. In our method, the sparsity of $\mathbf{W}$ can be either determined by a pre-defined pruning rate (See Section 3) or automatically adjusted by the stopping conditions in Section 3.5. We explore both effects in Section 4.

### 3.3 Discrimination-aware channel pruning

By introducing $P$ losses $\{\mathcal{L}_S^p\}_{p=1}^P$ to intermediate layers, the proposed discrimination-aware channel pruning (DCP) method is shown in Algorithm 1. Starting from a pre-trained model, DCP updates the model $M$ and performs channel pruning with $(P + 1)$ stages. Algorithm 1 is called discrimination-aware in the sense that an additional loss and the final loss are considered to fine-tune the model. Moreover, the additional loss will be used to select channels, as discussed below. In contrast to GoogLeNet [43] and DSN [22], in Algorithm 1, we do not use all the losses at the same time. In fact, at each stage we will consider two losses only, i.e., $\mathcal{L}_S^p$ and the final loss $\mathcal{L}_f$.

---

**Algorithm 1** Discrimination-aware channel pruning (DCP)

**Input:** Pre-trained model $M$, training data $\{\mathbf{x}_i, y_i\}_{i=1}^N$, and parameters $\{\kappa_l\}_{l=1}^L$.
**for** $p \in \{1, ..., P + 1\}$ **do**
    Construct loss $\mathcal{L}_S^p$ to layer $L_p$ as in Figure 1.
    Learn $\boldsymbol{\theta}$ and **Fine-tune** $M$ with $\mathcal{L}_S^p$ and $\mathcal{L}_f$.
    **for** $l \in \{L_{p-1} + 1, ..., L_p\}$ **do**
        Do **Channel Selection** for layer $l$ using Algorithm 2.
    **end for**
**end for**

---

**Algorithm 2** Greedy algorithm for channel selection

**Input:** Training data, model $M$, parameters $\kappa_l$, and $\epsilon$.
**Output:** Selected channel subset $\mathcal{A}$ and model parameters $\mathbf{W}_{\mathcal{A}}$.
Initialize $\mathcal{A} \leftarrow \emptyset$, and $t = 0$.
**while** (stopping conditions are not achieved) **do**
    Compute gradients of $\mathcal{L}$ w.r.t. $\mathbf{W}$: $\mathbf{G} = \partial \mathcal{L} / \partial \mathbf{W}$.
    Find the channel $k = \arg \max_{j \notin \mathcal{A}} \{||\mathbf{G}_j||_F\}$.
    Let $\mathcal{A} \leftarrow \mathcal{A} \cup \{k\}$.
    Solve Problem (8) to update $\mathbf{W}_{\mathcal{A}}$.
    Let $t \leftarrow t + 1$.
**end while**

---

At each stage of Algorithm 1, for example, in the $p$-th stage, we first construct the additional loss $\mathcal{L}_S^p$ and put them at layer $L_p$ (See Figure 1). After that, we learn the model parameters $\boldsymbol{\theta}$ w.r.t. $\mathcal{L}_S^p$ and fine-tune the model $M$ at the same time with both the additional loss $\mathcal{L}_S^p$ and the final loss $\mathcal{L}_f$. In the fine-tuning, all the parameters in $M$ will be updated.[4] Here, with the fine-tuning, the parameters regarding the additional loss can be well learned. Besides, fine-tuning is essential to compensate the accuracy loss from the previous pruning to suppress the accumulative error. After fine-tuning with $\mathcal{L}_S^p$ and $\mathcal{L}_f$, the discriminative power of layers $l \in \{L_{p-1} + 1, ..., L_p\}$ can be significantly improved. Then, we can perform channel selection for the layers in $\{L_{p-1} + 1, ..., L_p\}$.

### 3.4 Greedy algorithm for channel selection

Due to the $\ell_{2,0}$-norm constraint, directly optimizing Problem (7) is very difficult. To address this issue, following general greedy methods in [25, 2, 52, 45, 46], we propose a greedy algorithm to solve Problem (7). To be specific, we first remove all the channels and then select those channels that really contribute to the discriminative power of the deep networks. Let $\mathcal{A} \subset \{1, \dots, c\}$ be the index set of the selected channels, where $\mathcal{A}$ is empty at the beginning. As shown in Algorithm 2, the channel selection method can be implemented in two steps. First, we select the most important channels of input feature maps. At each iteration, we compute the gradients $\mathbf{G}_j = \partial \mathcal{L} / \partial \mathbf{W}_j$, where $\mathbf{W}_j$ denotes the parameters for the $j$-th input channel. We choose the channel $k = \arg \max_{j \notin \mathcal{A}} \{||\mathbf{G}_j||_F\}$ as an active channel and put $k$ into $\mathcal{A}$. Second, once $\mathcal{A}$ is determined, we optimize $\mathbf{W}$ w.r.t. the selected channels by minimizing the following problem:

$$\min_{\mathbf{W}} \quad \mathcal{L}(\mathbf{W}), \quad \text{s.t. } \mathbf{W}_{\mathcal{A}^c} = \mathbf{0}, \tag{8}$$

where $\mathbf{W}_{\mathcal{A}^c}$ denotes the submatrix indexed by $\mathcal{A}^c$ which is the complementary set of $\mathcal{A}$. Here, we apply stochastic gradient descent (SGD) to address the problem in Eq. (8), and update $\mathbf{W}_{\mathcal{A}}$ by

$$\mathbf{W}_{\mathcal{A}} \leftarrow \mathbf{W}_{\mathcal{A}} - \gamma \frac{\partial \mathcal{L}}{\partial \mathbf{W}_{\mathcal{A}}}, \tag{9}$$

where $\mathbf{W}_{\mathcal{A}}$ denotes the submatrix indexed by $\mathcal{A}$, and $\gamma$ denotes the learning rate.

Note that when optimizing Problem (8), $\mathbf{W}_{\mathcal{A}}$ is warm-started from the fine-tuned model $M$. As a result, the optimization can be completed very quickly. Moreover, since we only consider the model parameter $\mathbf{W}$ for one layer, we do not need to consider all data to do the optimization. To make a trade-off between the efficiency and performance, we sample a subset of images randomly from the training data for optimization.[5] Last, since we use SGD to update $\mathbf{W}_{\mathcal{A}}$, the learning rate $\gamma$ should be carefully adjusted to achieve an accurate solution. Then, the following stopping conditions can be applied, which will help to determine the number of channels to be selected.

### 3.5 Stopping conditions

Given a predefined parameter $\kappa_l$ in problem (7), Algorithm 2 will be stopped if $||\mathbf{W}||_{2,0} > \kappa_l$. However, in practice, the parameter $\kappa_l$ is hard to be determined. Since $\mathcal{L}$ is convex, $\mathcal{L}(\mathbf{W}^t)$ will monotonically decrease with iteration index $t$ in Algorithm 2. We can therefore adopt the following stopping condition:

$$|\mathcal{L}(\mathbf{W}^{t-1}) - \mathcal{L}(\mathbf{W}^t)|/\mathcal{L}(\mathbf{W}^0) \leq \epsilon, \tag{10}$$

where $\epsilon$ is a tolerance value. If the above condition is achieved, the algorithm is stopped, and the number of selected channels will be automatically determined, i.e., $||\mathbf{W}^t||_{2,0}$. An empirical study over the tolerance value $\epsilon$ is put in Section 5.3.

## 4 Experiments

In this section, we empirically evaluate the performance of DCP. Several state-of-the-art methods are adopted as the baselines, including ThiNet [28], Channel pruning (CP) [14] and Slimming [27]. Besides, to investigate the effectiveness of the proposed method, we include the following methods for study: **DCP:** DCP with a pre-defined pruning rate $\eta$. **DCP-Adapt:** We prune each layer with the stopping conditions in Section 3.5. **WM:** We shrink the width of a network by a fixed ratio and train it from scratch, which is known as width-multiplier [15]. **WM+:** Based on WM, we evenly insert additional losses to the network and train it from scratch. **Random DCP:** Relying on DCP, we randomly choose channels instead of using gradient-based strategy in Algorithm 2.

**Datasets.** We evaluate the performance of various methods on three datasets, including CIFAR-10 [20], ILSVRC-12 [4], and LFW [17]. CIFAR-10 consists of 50k training samples and 10k testing images with 10 classes. ILSVRC-12 contains 1.28 million training samples and 50k testing images for 1000 classes. LFW [17] contains 13,233 face images from 5,749 identities.

### 4.1 Implementation details

We implement the proposed method on PyTorch [32]. Based on the pre-trained model, we apply our method to select the informative channels. In practice, we decide the number of additional losses according to the depth of the network (See Section S4 in the supplementary material). Specifically, we insert 3 losses to ResNet-50 and ResNet-56, and 2 additional losses to VGGNet and ResNet-18.

We fine-tune the whole network with selected channels only. We use SGD with nesterov [30] for the optimization. The momentum and weight decay are set to 0.9 and 0.0001, respectively. We set $\lambda$ to 1.0 in our experiments by default. On CIFAR-10, we fine-tune 400 epochs using a mini-batch size of 128. The learning rate is initialized to 0.1 and divided by 10 at epoch 160 and 240. On ILSVRC-12, we fine-tune the network for 60 epochs with a mini-batch size of 256. The learning rate is started at 0.01 and divided by 10 at epoch 36, 48 and 54, respectively. The source code of our method can be found at `https://github.com/SCUT-AILab/DCP`.

### 4.2 Comparisons on CIFAR-10

We first prune ResNet-56 and VGGNet on CIFAR-10. The comparisons with several state-of-the-art methods are reported in Table 1. From the results, our method achieves the best performance under the same acceleration rate compared with the previous state-of-the-art. Moreover, with DCP-Adapt, our pruned VGGNet outperforms the pre-trained model by **0.58%** in testing error, and obtains **15.58×**

Table 1: Comparisons on CIFAR-10. "-" denotes that the results are not reported.

| Model | | ThiNet [28] | CP [14] | Sliming [27] | WM | WM+ | Random DCP | DCP | DCP-Adapt |
|---|---|---|---|---|---|---|---|---|---|
| VGGNet (Baseline 6.01%) | #Param. ↓ | 1.92× | 1.92× | 8.71× | 1.92× | 1.92× | 1.92× | 1.92× | **15.58×** |
| | #FLOPs ↓ | 2.00× | 2.00× | 2.04× | 2.00× | 2.00× | 2.00× | 2.00× | **2.86×** |
| | Err. gap (%) | +0.14 | +0.32 | +0.19 | +0.38 | +0.11 | +0.14 | -0.17 | **-0.58** |
| ResNet-56 (Baseline 6.20%) | #Param. ↓ | 1.97× | - | - | 1.97× | 1.97× | 1.97× | 1.97× | **3.37×** |
| | #FLOPs ↓ | 1.99× | 2× | - | 1.99× | 1.99× | 1.99× | 1.99× | 1.89× |
| | Err. gap (%) | +0.82 | +1.0 | - | +0.56 | +0.45 | +0.63 | **+0.31** | **-0.01** |

reduction in model size. Compared with random DCP, our proposed DCP reduces the performance degradation of VGGNet by 0.31%, which implies the effectiveness of the proposed channel selection strategy. Besides, we also observe that the inserted additional losses can bring performance gain to the networks. With additional losses, *WM+* of VGGNet outperforms *WM* by 0.27% in testing error. Nevertheless, our method shows much better performance than *WM+*. For example, our pruned VGGNet with DCP-Adapt outperforms *WM+* by **0.69%** in testing error.

**Pruning MobileNet v1 and MobileNet v2 on CIFAR-10.** We apply DCP to prune recently developed compact architectures, e.g., MobileNet v1 and MobileNet v2 , and evaluate the performance on CIFAR-10. We report the results in Table 2. With additional losses, *WM+* of MobileNet outperforms *WM* by 0.26% in testing error. However, our pruned models achieve 0.41% improvement over MobileNet v1 and 0.22% improvement over MobileNet v2 in testing error. Note that the **Random DCP** incurs performance degradation on both MobileNet v1 and MobileNet v2 by 0.30% and 0.57%, respectively.

Table 2: Performance of pruning 30% channels of MobileNet v1 and MobileNet v2 on CIFAR-10.

| Model | | WM | WM+ | Random DCP | DCP |
|---|---|---|---|---|---|
| MobileNet v1 (Baseline 6.04%) | #Param. ↓ | 1.43× | 1.43× | 1.43× | 1.43× |
| | #FLOPs ↓ | 1.75× | 1.75× | 1.75× | 1.75× |
| | Err. gap (%) | +0.48 | +0.22 | +0.30 | **-0.41** |
| MobileNet v2 (Baseline 5.53%) | #Param. ↓ | 1.31× | 1.31× | 1.31× | 1.31× |
| | #FLOPs ↓ | 1.36× | 1.36× | 1.36× | 1.36× |
| | Err. gap (%) | +0.45 | +0.40 | +0.57 | **-0.22** |

## 4.3 Comparisons on ILSVRC-12

To verify the effectiveness of the proposed method on large-scale datasets, we further apply our method on ResNet-50 to achieve 2× acceleration on ILSVRC-12. We report the single view evaluation in Table 3. Our method outperforms ThiNet [28] by **0.81%** and **0.51%** in top-1 and top-5 error, respectively. Compared with channel pruning [14], our pruned model achieves 0.79% improvement in top-5 error. Compared with *WM+*, which leads to 2.41% increase in top-1 error, our method only results in **1.06%** degradation in top-1 error.

Table 3: Comparisons on ILSVRC-12. The **top-1 and top-5 error (%)** of the pre-trained model are **23.99 and 7.07**, respectively. "-" denotes that the results are not reported.

| Model | | ThiNet [28] | CP [14] | WM | WM+ | DCP |
|---|---|---|---|---|---|---|
| ResNet-50 | #Param. ↓ | 2.06× | - | 2.06× | 2.06× | 2.06× |
| | #FLOPs ↓ | 2.25× | 2× | 2.25× | 2.25× | 2.25× |
| | Top-1 gap (%) | +1.87 | - | +2.81 | +2.41 | **+1.06** |
| | Top-5 gap (%) | +1.12 | +1.40 | +1.62 | +1.28 | **+0.61** |

## 4.4 Experiments on LFW

We further conduct experiments on LFW [17], which is a standard benchmark dataset for face recognition. We use CASIA-WebFace [50] (which consists of 494,414 face images from 10,575 individuals) for training. With the same settings in [26], we first train SphereNet-4 (which contains 4 convolutional layers) from scratch. And Then, we adopt our method to compress the pre-trained SphereNet model. Since the fully connected layer occupies 87.65% parameters of the model, we also prune the fully connected layer to reduce the model size.

Table 4: Comparisons of prediction accuracy, #Param. and #FLOPs on LFW. We report the ten-fold cross validation accuracy of different models.

| Method | FaceNet [37] | DeepFace [44] | VGG [31] | SphereNet-4 [26] | DCP (prune 50%) | DCP (prune 65%) |
|---|---|---|---|---|---|---|
| #Param. | 140M | 120M | 133M | 12.56M | 5.89M | 4.06M |
| #FLOPs | 1.6B | 19.3B | 11.3B | 164.61M | 45.15M | 24.16M |
| LFW acc. (%) | 99.63 | 97.35 | 99.13 | 98.20 | 98.30 | 98.02 |

We report the results in Table 4. With the pruning rate of 50%, our method speeds up SphereNet-4 for **3.66×** with **0.1%** improvement in ten-fold validation accuracy. Compared with huge networks, e.g., FaceNet [37], DeepFace [44], and VGG [31], our pruned model achieves comparable performance but has only **45.15M** FLOPs and **5.89M** parameters, which is sufficient to be deployed on embedded systems. Furthermore, pruning 65% channels in SphereNet-4 results in a more compact model, which requires only 24.16M FLOPs with the accuracy of 98.02% on LFW.

# 5 Ablation studies

## 5.1 Performance with different pruning rates

To study the effect of using different pruning rates $\eta$, we prune 30%, 50%, and 70% channels of ResNet-18 and ResNet-50, and evaluate the pruned models on ILSVRC-12. Experimental results are shown in Table 5. Here, we only report the performance under different pruning rates, while the detailed model complexity comparisons are provided in Section S8 in the supplementary material.

From Table 5, in general, performance of the pruned models goes worse with the increase of pruning rate. However, our pruned ResNet-50 with pruning rate of 30% outperforms the pre-trained model, with **0.39%** and **0.14%** reduction in top-1 and top-5 error, respectively. Besides, the performance degradation of ResNet-50 is smaller than that of ResNet-18 with the same pruning rate. For example, when pruning 50% of the channels, while it only leads to 1.06% increase in top-1 error for ResNet-50, it results in 2.29% increase of top-1 error for ResNet-18. One possible reason is that, compared to ResNet-18, ResNet-50 is more redundant with more parameters, thus it is easier to be pruned.

Table 5: Comparisons on ResNet-18 and ResNet-50 with different pruning rates. We report the top-1 and top-5 error (%) on ILSVRC-12.

| Network | $\eta$ | Top-1/Top5 err. |
|---|---|---|
| ResNet-18 | 0% (baseline) | 30.36/11.02 |
|  | 30% | **30.79/11.14** |
|  | 50% | 32.65/12.40 |
|  | 70% | 35.88/14.32 |
| ResNet-50 | 0% (baseline) | 23.99/7.07 |
|  | 30% | **23.60/6.93** |
|  | 50% | 25.05/7.68 |
|  | 70% | 27.25/8.87 |

Table 6: Pruning results on ResNet-56 with different $\lambda$ on CIFAR-10.

| $\lambda$ | Training err. | Testing err. |
|---|---|---|
| 0 ($\mathcal{L}_M$ only) | 7.96 | 12.24 |
| 0.001 | 7.61 | 11.89 |
| 0.005 | 6.86 | 11.24 |
| 0.01 | 6.36 | 11.00 |
| 0.05 | 4.18 | 9.74 |
| 0.1 | 3.43 | 8.87 |
| 0.5 | 2.17 | 8.11 |
| 1.0 | **2.10** | **7.84** |
| 1.0 ($\mathcal{L}_S$ only) | 2.82 | 8.28 |

## 5.2 Effect of the trade-off parameter $\lambda$

We prune 30% channels of ResNet-56 on CIFAR-10 with different $\lambda$. We report the training error and testing error without fine-tuning in Table 6. From the table, the performance of the pruned model improves with increasing $\lambda$. Here, a larger $\lambda$ implies that we put more emphasis on the additional loss (See Equation (6)). This demonstrates the effectiveness of discrimination-aware strategy for channel selection. It is worth mentioning that both the reconstruction error and the cross-entropy loss contribute to better performance of the pruned model, which strongly supports the motivation to select the important channels by $\mathcal{L}_S$ and $\mathcal{L}_M$. After all, as the network achieves the best result when $\lambda$ is set to 1.0, we use this value to initialize $\lambda$ in our experiments.

## 5.3 Effect of the stopping condition

To explore the effect of stopping condition discussed in Section 3.5, we test different tolerance value $\epsilon$ in the condition. Here, we prune VGGNet on CIFAR-10 with $\epsilon \in \{0.1, 0.01, 0.001\}$. Experimental results are shown in Table 7. In general, a smaller $\epsilon$ will lead to more rigorous stopping condition and hence more channels will be selected. As a result, the performance of the pruned model is improved with the decrease of $\epsilon$. This experiment demonstrates the usefulness and effectiveness of the stopping condition for automatically determining the pruning rate.

Table 7: Effect of $\epsilon$ for channel selection. We prune VGGNet and report the testing error on CIFAR-10. The testing error of baseline VGGNet is 6.01%.

| Loss | $\epsilon$ | Testing err. (%) | #Param. $\downarrow$ | #FLOPs $\downarrow$ |
|---|---|---|---|---|
| | 0.1 | 12.68 | **152.25×** | 27.39× |
| $\mathcal{L}$ | 0.01 | 6.63 | 31.28× | 5.35× |
| | 0.001 | **5.43** | 15.58× | 2.86× |

## 5.4 Visualization of feature maps

We visualize the feature maps w.r.t. the pruned/selected channels of the first block (i.e., res-2a) in ResNet-18 in Figure 2. From the results, we observe that feature maps of the pruned channels (See Figure 2(b)) are less informative compared to those of the selected ones (See Figure 2(c)). It proves that the proposed DCP selects the channels with strong discriminative power for the network. More visualization results can be found in Section S10 in the supplementary material.

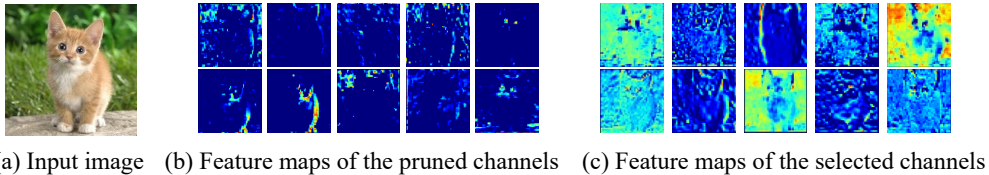

(a) Input image  (b) Feature maps of the pruned channels  (c) Feature maps of the selected channels

Figure 2: Visualization of the feature maps of the pruned/selected channels of res-2a in ResNet-18.

## 6 Conclusion

In this paper, we have proposed a discrimination-aware channel pruning method for the compression of deep neural networks. We formulate the channel pruning/selection problem as a sparsity-induced optimization problem by considering both reconstruction error and channel discrimination power. Moreover, we propose a greedy algorithm to solve the optimization problem. Experimental results on benchmark datasets show that the proposed method outperforms several state-of-the-art methods by a large margin with the same pruning rate. Our DCP method provides an effective way to obtain more compact networks. For those compact network designs such as MobileNet v1&v2, DCP can still improve their performance by removing redundant channels. In particular for MobileNet v2, DCP improves it by reducing 30% of channels on CIFAR-10. In the future, we will incorporate the computational cost per layer into the optimization, and combine our method with other model compression strategies (such as quantization) to further reduce the model size and inference cost.

## Acknowledgements

This work was supported by National Natural Science Foundation of China (NSFC) (61876208, 61502177 and 61602185), Recruitment Program for Young Professionals, Guangdong Provincial Scientific and Technological funds (2017B090901008, 2017A010101011, 2017B090910005), Fundamental Research Funds for the Central Universities D2172480, Pearl River S&T Nova Program of Guangzhou 201806010081, CCF-Tencent Open Research Fund RAGR20170105, and Program for Guangdong Introducing Innovative and Enterpreneurial Teams 2017ZT07X183.

## Footnotes

[3]The proof can be found in Section S1 in the supplementary material.

[4]The details of fine-tuning algorithm is put in Section S2 in the supplementary material.

[5]We study the effect of the number of samples in Section S5 in the supplementary material.

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
