[Supplementary Material]

# Supplementary Material: Discrimination-aware Channel Pruning for Deep Neural Networks

**Zhuangwei Zhuang**[1*], **Mingkui Tan**[1*†], **Bohan Zhuang**[2*], **Jing Liu**[1*],
**Yong Guo**[1], **Qingyao Wu**[1], **Junzhou Huang**[3,4], **Jinhui Zhu**[1†]
[1]South China University of Technology, [2]The University of Adelaide,
[3]University of Texas at Arlington, [4]Tencent AI Lab
{z.zhuangwei, seliujing, guo.yong}@mail.scut.edu.cn, jzhuang@uta.edu
{mingkuitan, qyw, csjhzhu}@scut.edu.cn, bohan.zhuang@adelaide.edu.au

We organize our supplementary material as follows. In Section S1, we give some theoretical analysis on the loss function. In Section S2, we introduce the details of fine-tuning algorithm in DCP. Then, in Section S3, we discuss the effect of pruning each individual block in ResNet-18. We explore the number of additional losses in Section S4. We explore the effect of the number of samples on channel selection in Section S5. We study the influence on the quality of pre-trained models in Section S6. In Section S7, we apply our method to prune MobileNet v1 and MobileNet v2 on ILSVRC-12. We discuss the model complexities of the pruned models in Section S8, and report the detailed structure of the pruned VGGNet with DCP-Adapt in Section S9. We provide more visualization results of the feature maps w.r.t. the pruned/selected channels in Section S10.

## S1 Convexity of the loss function

In this section, we analyze the property of the loss function.

**Proposition 1 (Convexity of the loss function)** *Let* $\mathbf{W}$ *be the model parameters of a considered layer. Given the mean square loss and the cross-entropy loss defined in Eqs. (5) and (3), then the joint loss function* $\mathcal{L}(\mathbf{W})$ *is convex w.r.t.* $\mathbf{W}$.

**Proof 1** *The mean square loss (3) w.r.t.* $\mathbf{W}$ *is convex because* $\mathbf{O}_{i,j,:,:}$ *is linear w.r.t.* $\mathbf{W}$. *Without loss of generality, we consider the cross-entropy of binary classification, it can be extend to multi-classification, i.e.,*

$$\mathcal{L}_S^p(\mathbf{W}) = \sum_{i=1}^N y^i \left[ -\log \left( h_{\boldsymbol{\theta}} \left( \mathbf{F}^{(p,i)} \right) \right) \right] + (1 - y^i) \left[ -\log \left( 1 - h_{\boldsymbol{\theta}} \left( \mathbf{F}^{(p,i)} \right) \right) \right],$$

*where* $h_{\boldsymbol{\theta}} \left( \mathbf{F}^{(p,i)} \right) = \frac{1}{1+e^{-\boldsymbol{\theta}^{\mathsf{T}} \mathbf{F}^{(p,i)}}}$, $\mathbf{F}^p(\mathbf{W}) = \text{AvgPooling}(\text{ReLU}(\text{BN}(\mathbf{O}^p)))$ *and* $\mathbf{O}_{i,j,:,:}^p$ *is linear w.r.t.* $\mathbf{W}$. *Here, we assume* $\mathbf{F}^{(p,i)}$ *and* $\mathbf{W}$ *are vectors. The loss function* $\mathcal{L}_S^p(\mathbf{W})$ *is convex w.r.t.* $\mathbf{W}$ *as long as* $-\log \left( h_{\boldsymbol{\theta}} \left( \mathbf{F}^{(p,i)} \right) \right)$ *and* $-\log \left( 1 - h_{\boldsymbol{\theta}} \left( \mathbf{F}^{(p,i)} \right) \right)$ *are convex w.r.t.* $\mathbf{W}$. *First, we calculate the derivative of the former, we have*

$$\nabla_{\mathbf{F}^{(p,i)}} \left[ -\log \left( h_{\boldsymbol{\theta}} \left( \mathbf{F}^{(p,i)} \right) \right) \right] = \nabla_{\mathbf{F}^{(p,i)}} \left[ \log \left( 1 + e^{-\boldsymbol{\theta}^{\mathsf{T}} \mathbf{F}^{(p,i)}} \right) \right] = \left( h_{\boldsymbol{\theta}} \left( \mathbf{F}^{(p,i)} \right) - 1 \right) \boldsymbol{\theta}$$

*and*

$$\begin{aligned} \nabla_{\mathbf{F}^{(p,i)}}^2 \left[ -\log \left( h_{\boldsymbol{\theta}} \left( \mathbf{F}^{(p,i)} \right) \right) \right] &= \nabla_{\mathbf{F}^{(p,i)}} \left[ \left( h_{\boldsymbol{\theta}} \left( \mathbf{F}^{(p,i)} \right) - 1 \right) \boldsymbol{\theta} \right] \\ &= h_{\boldsymbol{\theta}} \left( \mathbf{F}^{(p,i)} \right) \left( 1 - h_{\boldsymbol{\theta}} \left( \mathbf{F}^{(p,i)} \right) \right) \boldsymbol{\theta} \boldsymbol{\theta}^{\mathsf{T}}. \end{aligned} \tag{S1}$$

---

[*]Authors contributed equally.
[†]Corresponding author.

*Using chain rule, the derivative w.r.t.* $\mathbf{W}$ *is*

$$\nabla_{\mathbf{W}}\left[-\log\left(h_{\boldsymbol{\theta}}\left(\mathbf{F}^{(p,i)}\right)\right)\right] = \left(\nabla_{\mathbf{W}}\mathbf{F}^{(p,i)}\right)\nabla_{\mathbf{F}^{(p,i)}}\left[-\log\left(h_{\boldsymbol{\theta}}\left(\mathbf{F}^{(p,i)}\right)\right)\right]$$

$$= \left(\nabla_{\mathbf{W}}\mathbf{F}^{(p,i)}\right)\left(h_{\boldsymbol{\theta}}\left(\mathbf{F}^{(p,i)}\right)-1\right)\boldsymbol{\theta},$$

*then the hessian matrix is*

$$\nabla_{\mathbf{W}}^2\left[-\log\left(h_{\boldsymbol{\theta}}\left(\mathbf{F}^{(p,i)}\right)\right)\right]$$

$$=\nabla_{\mathbf{W}}\left[\left(\nabla_{\mathbf{W}}\mathbf{F}^{(p,i)}\right)\left(h_{\boldsymbol{\theta}}\left(\mathbf{F}^{(p,i)}\right)-1\right)\boldsymbol{\theta}\right]$$

$$=\nabla_{\mathbf{W}}^2\mathbf{F}^{(p,i)}\left(h_{\boldsymbol{\theta}}\left(\mathbf{F}^{(p,i)}\right)-1\right)\boldsymbol{\theta}+\nabla_{\mathbf{W}}\left[\left(h_{\boldsymbol{\theta}}\left(\mathbf{F}^{(p,i)}\right)-1\right)\boldsymbol{\theta}\right]\left(\nabla_{\mathbf{W}}\mathbf{F}^{(p,i)}\right)^{\mathsf{T}}$$

$$=\nabla_{\mathbf{W}}\left[\left(h_{\boldsymbol{\theta}}\left(\mathbf{F}^{(p,i)}\right)-1\right)\boldsymbol{\theta}\right]\left(\nabla_{\mathbf{W}}\mathbf{F}^{(p,i)}\right)^{\mathsf{T}}$$

$$=\left(\nabla_{\mathbf{W}}\mathbf{F}^{(p,i)}\right)\nabla_{\mathbf{F}^{(p,i)}}\left[\left(h_{\boldsymbol{\theta}}\left(\mathbf{F}^{(p,i)}\right)-1\right)\boldsymbol{\theta}\right]\left(\nabla_{\mathbf{W}}\mathbf{F}^{(p,i)}\right)^{\mathsf{T}}$$

$$=\left(\nabla_{\mathbf{W}}\mathbf{F}^{(p,i)}\right)h_{\boldsymbol{\theta}}\left(\mathbf{F}^{(p,i)}\right)\left(1-h_{\boldsymbol{\theta}}\left(\mathbf{F}^{(p,i)}\right)\right)\boldsymbol{\theta}\boldsymbol{\theta}^{\mathsf{T}}\left(\nabla_{\mathbf{W}}\mathbf{F}^{(p,i)}\right)^{\mathsf{T}}.$$

*The third equation is hold by the fact that* $\nabla_{\mathbf{W}}^2\mathbf{F}^{(p,i)}=0$ *and the last equation is follows by Eq. (S1).*
*Therefore, the hessian matrix is semi-definite because* $h_{\boldsymbol{\theta}}\left(\mathbf{F}^{(p,i)}\right)\geq 0$, $1-h_{\boldsymbol{\theta}}\left(\mathbf{F}^{(p,i)}\right)\geq 0$ *and*

$$\mathbf{z}^{\mathsf{T}}\nabla_{\mathbf{W}}^2\left[-\log\left(h_{\boldsymbol{\theta}}\left(\mathbf{F}^{(p,i)}\right)\right)\right]\mathbf{z}=h_{\boldsymbol{\theta}}\left(\mathbf{F}^{(p,i)}\right)\left(1-h_{\boldsymbol{\theta}}\left(\mathbf{F}^{(p,i)}\right)\right)\left(\mathbf{z}^{\mathsf{T}}\nabla_{\mathbf{W}}\mathbf{F}^{(p,i)}\boldsymbol{\theta}\right)^2\geq 0,\ \forall\mathbf{z}.$$

*Similarly, the hessian matrix of the latter one of loss function is also semi-definite. Therefore, the joint loss function* $\mathcal{L}(\mathbf{W})$ *is convex w.r.t.* $\mathbf{W}$.

## S2 Details of fine-tuning algorithm in DCP

Let $L_p$ be the position of the inserted output in the $p$-th stage. $\mathbf{W}$ denotes the model parameters. We apply forward propagation once, and compute the additional loss $\mathcal{L}_S^p$ and the final loss $\mathcal{L}_f$. Then, we compute the gradients of $\mathcal{L}_S^p$ w.r.t. $\mathbf{W}$, and update $\mathbf{W}$ by

$$\mathbf{W}\leftarrow\mathbf{W}-\gamma\frac{\partial\mathcal{L}_S^p}{\partial\mathbf{W}},\tag{S2}$$

where $\gamma$ denotes the learning rate.

Based on the last $\mathbf{W}$, we compute the gradient of $\mathcal{L}_f$ w.r.t. $\mathbf{W}$, and update $\mathbf{W}$ by

$$\mathbf{W}\leftarrow\mathbf{W}-\gamma\frac{\partial\mathcal{L}_f}{\partial\mathbf{W}}.\tag{S3}$$

The fine-tuning algorithm in DCP is shown in Algorithm S1.

---

**Algorithm S1** Fine-tuning Algorithm in DCP

---

**Input:** Position of the inserted output $L_p$, model parameters $\mathbf{W}$, the number of fine-tuning iteration $T$, learning rate $\gamma$, decay of learning rate $\tau$.
**for** Iteration $t=1$ to $T$ **do**
  Randomly choose a mini-batch of samples from the training set.
  Compute gradient of $\mathcal{L}_S^p$ w.r.t. $\mathbf{W}$: $\frac{\partial\mathcal{L}_S^p}{\partial\mathbf{W}}$.
  Update $\mathbf{W}$ using Eq. (S2).
  Compute gradient of $\mathcal{L}_f$ w.r.t. $\mathbf{W}$: $\frac{\partial\mathcal{L}_f}{\partial\mathbf{W}}$.
  Update $\mathbf{W}$ using Eq. (S3).
  $\gamma\leftarrow\tau\gamma$.
**end for**

---

## S3  Channel pruning in a single block

To evaluate the effectiveness of our method on pruning channels in a single block, we apply our method to each block in ResNet-18 separately. We implement the algorithms in ThiNet [4], APoZ [2] and weight sum [3], and compare the performance on ILSVRC-12 with pruning 30% channels in the network. As shown in Figure S1, our method outperforms the strategies of APoZ and weight sum significantly. Compared with ThiNet, our method achieves lower degradation of performance under the same pruning rate, especially in the deeper layers.

Figure S1: Pruning different blocks in ResNet-18. We report the increased top-1 error on ILSVRC-12.

## S4  Exploring the number of additional losses

To study the effect of the number of additional losses, we prune 50% channels from ResNet-56 for $2\times$ acceleration on CIFAR-10. As shown in Table S1, adding too many losses may lead to little gain in performance but incur significant increase of computational cost. Heuristically, we find that adding losses every 5-10 layers is sufficient to make a good trade-off between accuracy and complexity.

Table S1: Effect on the number of additional losses over ResNet-56 for $2\times$ acceleration on CIFAR-10.

| #additional losses | 3 | 5 | 7 | 9 |
|---|---|---|---|---|
| Error gap (%) | +0.31 | +0.27 | +0.21 | +0.20 |

## S5  Exploring the number of samples

To study the influence of the number of samples on channel selection, we prune 30% channels from ResNet-18 on ILSVRC-12 with different number of samples, i.e., from 10 to 100k. Experimental results are shown in Figure S2.

Figure S2: Testing error on ILSVRC-12 with different number of samples for channel selection.

In general, with more samples for channel selection, the performance degradation of the pruned model can be further reduced. However, it also leads to more expensive computation cost. To make a trade-off between performance and efficiency, we use 10k samples in our experiments for ILSVRC-12. For small datasets like CIFAR-10, we use the whole training set for channel selection.

## S6 Influence on the quality of pre-trained models

To explore the influence on the quality of pre-trained models, we use intermediate models at epochs {120, 240, 400} from ResNet-56 for $2\times$ acceleration as pre-trained models, which have different quality. From the results in Table S2, DCP shows small sentity to the quality of pre-trained models. Moreover, given models of the same quality, DCP steadily outperforms the other two methods.

Table S2: Influence of pre-trained model quality over ResNet-56 for $2\times$ acceleration on CIFAR-10.

| Epochs (baseline error) | ThiNet | Channel pruning | DCP |
|---|---|---|---|
| 120 (10.57%) | +1.22 | +1.39 | +0.39 |
| 240 (6.51%) | +0.92 | +1.02 | +0.36 |
| 400 (6.20%) | +0.82 | +1.00 | +0.31 |

## S7 Pruning MobileNet v1 and MobileNet v2 on ILSVRC-12

We apply our DCP method to do channel pruning based on MobileNet v1 and MobileNet v2 on ILSVRC-12. The results are reported in Table S3. Our method outperforms ThiNet [4] in top-1 error by **0.75%** and **0.47%** on MobileNet v1 and MobileNet v2, respectively.

Table S3: Comparisons of MobileNet v1 and MobileNet v2 on ILSVRC-12. "-" denotes that the results are not reported.

| Model | | ThiNet [4] | WM [1, 5] | DCP |
|---|---|---|---|---|
| | #Param. ↓ | 2.00× | 2.00× | 2.00× |
| MobileNet v1 | #FLOPs ↓ | 3.49× | 3.49× | 3.49× |
| (Baseline 31.15%) | Top-1 gap (%) | +4.67 | +6.90 | **+3.92** |
| | Top-5 gap (%) | +3.36 | - | **+2.71** |
| | #Param. ↓ | 1.35× | 1.35× | 1.35× |
| MobileNet v2 | #FLOPs ↓ | 1.81× | 1.81× | 1.81× |
| (Baseline 29.89%) | Top-1 gap (%) | +6.36 | +6.40 | **+5.89** |
| | Top-5 gap (%) | +3.67 | +4.60 | +3.77 |

## S8 Complexity of the pruned models

We report the model complexities of our pruned models w.r.t. different pruning rates in Table S4 and Table S5. We evaluate the forward/backward running time on CPU/GPU. We perform the evaluations on a workstation with two Intel Xeon-E2630v3 CPU and a NVIDIA TitanX GPU. The mini-batch size is set to 32. Normally, as channels pruning removes the whole channels and related filters directly, it reduces the number of parameters and FLOPs of the network, resulting in acceleration in forward and backward propagation. We also report the speedup of running time w.r.t. the pruned ResNet18 and ResNet50 under different pruning rates in Figure S3. The speedup on CPU is higher than GPU. Although the pruned ResNet-50 with the pruning rate of 50% has similar computational cost to the ResNet-18, it requires $2.38\times$ GPU running time and $1.59\times$ CPU running time to ResNet-18. One possible reason is that wider networks are more efficient than deeper networks, as it can be efficiently paralleled on both CPU and GPU.

Table S4: Model complexity of the pruned ResNet-18 and ResNet-50 on ILSVRC-2012. f./b. indicates the forward/backward time tested on one NVIDIA TitanX GPU or two Intel Xeon-E2630v3 CPU with a mini-batch size of 32.

| Network | Prune rate (%) | #Param. | #FLOPs | GPU time (ms) | | CPU time (s) | |
|---------|---------------|---------|--------|---------------|-------|--------------|-------|
| | | | | f./b. | Total | f./b. | Total |
| ResNet-18 | 0 | $1.17 \times 10^7$ | $1.81 \times 10^9$ | 14.41/33.39 | 47.80 | 2.78/3.60 | 6.38 |
| | 30 | $8.41 \times 10^6$ | $1.32 \times 10^9$ | 13.10/29.40 | 42.50 | 2.18/2.70 | 4.88 |
| | 50 | $6.19 \times 10^6$ | $9.76 \times 10^8$ | 10.68/25.22 | 35.90 | 1.74/2.18 | 3.92 |
| | 70 | $4.01 \times 10^6$ | $6.49 \times 10^8$ | 9.74/22.60 | 32.34 | 1.46/1.75 | 3.21 |
| ResNet-50 | 0 | $2.56 \times 10^7$ | $4.09 \times 10^9$ | 49.97/109.69 | 159.66 | 6.86/9.19 | 16.05 |
| | 30 | $1.70 \times 10^7$ | $2.63 \times 10^9$ | 43.86/96.88 | 140.74 | 5.48/6.89 | 12.37 |
| | 50 | $1.24 \times 10^7$ | $1.82 \times 10^9$ | 35.48/78.23 | 113.71 | 4.42/5.74 | 10.16 |
| | 70 | $8.71 \times 10^6$ | $1.18 \times 10^9$ | 33.28/72.46 | 105.74 | 3.33/4.46 | 7.79 |

Table S5: Model complexity of the pruned ResNet-56 and VGGNet on CIFAR-10.

| Network | Prune rate (%) | #Param. | #FLOPs |
|---------|---------------|---------|--------|
| ResNet-56 | 0 | $8.56 \times 10^5$ | $1.26 \times 10^8$ |
| | 30 | $6.08 \times 10^5$ | $9.13 \times 10^7$ |
| | 50 | $4.31 \times 10^5$ | $6.32 \times 10^7$ |
| | 70 | $2.71 \times 10^5$ | $3.98 \times 10^7$ |
| VGGNet | 0 | $2.00 \times 10^7$ | $3.98 \times 10^8$ |
| | 30 | $1.04 \times 10^7$ | $1.99 \times 10^8$ |
| | 40 | $7.83 \times 10^6$ | $1.47 \times 10^8$ |
| MobileNet | 0 | $3.22 \times 10^6$ | $2.13 \times 10^8$ |
| | 30 | $2.25 \times 10^6$ | $1.22 \times 10^8$ |

Figure S3: Speedup of running time w.r.t. ResNet18 and ResNet50 under different pruning rates.

Figure S4: Pruning rates w.r.t. each layer in VGGNet. We measure the pruning rate by the ratio of pruned input channels.

## S9  Detailed structure of the pruned VGGNet

We show the detailed structure and pruning rate of a pruned VGGNet obtained from *DCP-Adapt* on CIFAR-10 dataset in Table S6 and Figure S4, respectively. Compared with the original network, the pruned VGGNet has lower layer complexities, especially in the deep layer.

Table S6: Detailed structure of the pruned VGGNet obtained from DCP-Adapt. "#Channel" and "#Channel$^*$" indicates the number of input channels of convolutional layers in the original VGGNet (testing error 6.01%) and a pruned VGGNet (testing error 5.43%) respectively.

| Layer | #Channel | #Channel$^*$ | Pruning rate (%) |
|-------|----------|-----------|------------------|
| conv1-1 | 3 | 3 | 0 |
| conv1-2 | 64 | 56 | 12.50 |
| conv2-1 | 64 | 64 | 0 |
| conv2-2 | 128 | 128 | 0 |
| conv3-1 | 128 | 115 | 10.16 |
| conv3-2 | 256 | 199 | 22.27 |
| conv3-3 | 256 | 177 | 30.86 |
| conv3-4 | 256 | 123 | 51.95 |
| conv4-1 | 256 | 52 | 79.69 |
| conv4-2 | 512 | 59 | 88.48 |
| conv4-3 | 512 | 46 | 91.02 |
| conv4-4 | 512 | 31 | 93.94 |
| conv4-5 | 512 | 37 | 92.77 |
| conv4-6 | 512 | 37 | 92.77 |
| conv4-7 | 512 | 44 | 91.40 |
| conv4-8 | 512 | 31 | 93.94 |

## S10  More visualization of feature maps

In Section 5.4, we have revealed the visualization of feature maps w.r.t. the pruned/selected channels. In this section, we provide more results of feature maps w.r.t. different input images, which are shown in Figure S5. According to Figure S5, the selected channels contain much more information than the pruned ones.

(a) Input image  (b) Feature maps of the pruned channels  (c) Feature maps of the selected channels

(a) Input image    (b) Feature maps of the pruned channels    (c) Feature maps of the selected channels

Figure S5: Visualization of feature maps w.r.t. the pruned/selected channels of res-2a in ResNet-18.