[Reviews · NeurIPS 2018]

Reviewer 1



The paper presents an approach for channel pruning in deep neural networks. The authors propose to prune channels based on their "discriminative power" -- this is achieved by introducing additional per-layer losses, and performing channel selection and network parameter optimization in an iterative fashion. Experimental results on 3 datasets (CIFAR-10, ILSVRC-12, LFW) show that the method compares favorably with respect to the state of the art. Overall, the paper is well written, and explanations are clear. The Supp. Material provides many further details / experiments. However, some parts of the text could be made clearer (e.g. : lines 47-53 pg. 2 became clear to me only after reading Sec. 3; notation in lines 135-140 pg. 4 could be simplified). The experimental results show the effectiveness of the proposed approach, which seems to bring an original contribution. However, for some technical choices, it would be useful to get a better intuition/motivation. Did the authors experiment with different losses than (5)? Could it happen that some layers capture high-level features, which are difficult to evaluate based on the cross-entropy loss? It would be interesting to give insights about how the approach could be adapted to work for models trained with different (final) losses. Furthermore, it is unclear to me how the layers L_p (the ones with added losses) are selected. Is there any heuristic to decide how many per-layer losses should be added, and to which layers? Finally, it would be useful to discuss how much computational effort/time the approach adds at training time. Other notes / questions: - It would be interesting to discuss in more detail the reasons for the improvement observed on ILSVRC-12 after pruning. In some cases, did the authors observe better generalization capabilities for the model? - It would be interesting to discuss if there are architectures for which the proposed approach is more/less suitable. - Does the quality of the pre-trained model heavily affect performances? - Sec. 3.2: When discussing limitations of reconstruction methods, it would be useful to add references supporting the claims. - When using acronyms, it would be better to write also the entire name, at the first occurrence (e.g. Stochastic Gradient Descent - SGD). - References: [10] appeared at ICCV, not CVPR. - Line 197, pg. 6: < instead of > ?

Reviewer 2



This paper present new channel pruning approach for CNN compression. (Pros) The paper is well-written and well-illustrated; hence it was easy to understand the authors’ proposal. It reports the minimum accuracy loss compare with existing channel pruning approach. (Cons) However, I am somewhat skeptical on this paper because recently proposed compact CNN (e.g. ShuffleNet, MobileNet v2) can achieve higher accuracy with smaller model size and MAC Gops than channel-pruned model in this paper. In the experiment, relatively heavy networks are used for model compression. Instead of these heavy network, If the proposed approach can squeeze the compact network successfully, then impact of this paper will be increased. Although there is an experimental result for MobileNet on CIFAR-10 in supplementary material, I think it is not enough. Results on ILSVRC-12 are needed. Another skeptical point is that the fine-tuning process is too long compare to other methods – 60 epochs for ILSVRC-12 (vs 13 or 10 epochs in ThiNet). It is not clear the small accuracy loss is due to the proposed pruning approach or just long training.

Reviewer 3



This paper proposes a method (DCP) to prune away channals in CNN by introducing additional losses to the intermediate layers. Original model weights are fine-tuned with the additional weights and channels are selected in a greedy manner layer by layer. Extensive experiments show improved performance against the state-of-the-art methods. The authors made the paper easy to following by clearly presenting the motivation, contribution and the method itself. While adding discrimative losses to intermediate layers of neural nets is not entirely new, the proposed method does demonstrate reasonable amount of novelty to obtain good empirical performance. I feel the strength of the paper lies in the extensive experiments; comparison to existing methods seems exhaustive and the ablative analysis appears robust. Some questions: * Why are the gradients with respect to the weights used in greedy channel selection, not the weights themselves? Did you try channel selection based on weights? * Is there a rule of thumb to determine the number of channels to prune for each layer? E.g. prune fewer channels in the first layer and prune more aggresively in the later layers?